# Physicochemical and Biological Characterization of Ti6Al4V Particles Obtained by Implantoplasty: An In Vitro Study. Part I

**DOI:** 10.3390/ma14216507

**Published:** 2021-10-29

**Authors:** Jorge Toledano-Serrabona, Francisco Javier Gil, Octavi Camps-Font, Eduard Valmaseda-Castellón, Cosme Gay-Escoda, Maria Ángeles Sánchez-Garcés

**Affiliations:** 1Bellvitge Biomedical Research Institute (IDIBELL), Department of Oral Surgery and Implantology, Faculty of Medicine and Health Sciences, University of Barcelona, 08907 Barcelona, Spain; jorgetoledano25@gmail.com (J.T.-S.); ocamps@ub.edu (O.C.-F.); eduardvalmaseda@ub.edu (E.V.-C.); cgay@ub.edu (C.G.-E.); 2Bioengineering Institute of Technology, International University of Catalonia, 08195 Barcelona, Spain; 3Faculty of Dentistry, International University of Catalonia, 08195 Barcelona, Spain

**Keywords:** dental implant, peri-implantitis, titanium alloy, implantoplasty

## Abstract

Implantoplasty is a mechanical decontamination technique that consists of polishing the supra-osseous component of the dental implant with peri-implantitis. This technique releases metal particles in the form of metal swarf and dust into the peri-implant environment. In the present in vitro study, the following physicochemical characterization tests were carried out: specific surface area, granulometry, contact angle, crystalline structure, morphology, and ion release. Besides, cytotoxicity was in turn evaluated by determining the fibroblastic and osteoblastic cell viability. As a result, the metal debris obtained by implantoplasty presented an equivalent diameter value of 159 µm (range 6–1850 µm) and a specific surface area of 0.3 m^2^/g on average. The particle had a plate-like shape of different sizes. The release of vanadium ions in Hank’s solution at 37 °C showed no signs of stabilization and was greater than that of titanium and aluminum ions, which means that the alloy suffers from a degradation. The particles exhibited cytotoxic effects upon human osteoblastic and fibroblastic cells in the whole extract. In conclusion, metal debris released by implantoplasty showed different sizes, surface structures and shapes. Vanadium ion levels were higher than that those of the other metal ions, and cell viability assays showed that these particles produce a significant loss of cytocompatibility on osteoblasts and fibroblasts, which means that the main cells of the peri-implant tissues might be injured.

## 1. Introduction

Titanium (Ti) is a transition metal with a metallic color and high chemical stability. Several metals, such as aluminum, vanadium, molybdenum, etc., can be added to Ti to improve its mechanical properties. Commercially pure (c.p.) Ti or Ti alloy are commonly used in dental implants and orthopedics (joint replacement) due to their corrosion resistance, biocompatibility, and mechanical properties [1]. Among the different Ti alloys, Ti6Al4V is the most commonly used and is applied in a wide range of applications such as dental implants, orthopedic prostheses, chemical industry, aerospace industry, etc. This alloy is easy to manufacture and has excellent corrosion resistance and biocompatibility as a stable, continuous, and inert oxide layer is established [2]. However, there are some concerns related to the usage of Ti6Al4V in the long term. The first is that several studies report that shear or wear debris from this alloy can trigger an inflammatory reaction that can lead to osteolysis and loss of the prosthesis. Another issue of this alloy is of potential V ion releasing, which can be toxic to the human body [3,4]. Besides, this alloy has some disadvantages such as low surface hardness, low fatigue strength, low wear properties and high coefficient of friction [5]. Taking into consideration the above-mentioned disadvantages, other alloys such as Ti-Zr, Ti-6Al-7Nb or others have been proposed to overcome these limitations, but there is still not enough long-term data [3,6].

Since Brånemark first described osseointegration [7], titanium (Ti) dental implants have proven to be a predictable therapeutic option for rehabilitating edentulous or partially edentulous patients. Although the success rate of dental implants exceeds 94% after 10 years [8], biological, mechanical, and esthetic complications may occur after oral rehabilitation [9].

Mucositis and peri-implantitis are bacterial plaque-induced inflammatory conditions that affect the tissues surrounding dental implants [10]. Dissimilar to mucositis, peri-implantitis is characterized by the presence of inflammatory signs with progressive marginal bone loss [11]. The frequency of peri-implant disease is high, with a reported prevalence of peri-implantitis of 12.8% at implant level and 18.5% at patient level [12].

Different surgical approaches, such as access surgery, resective surgery, regenerative surgery, or a combination of these, have been suggested for the treatment of peri-implantitis [13]. After resective surgery, the rough surface of the implant may be exposed to the oral environment. In these cases, implantoplasty (IP) has been described to decontaminate the supracrestal component of the implant, hinder bacterial adhesion and facilitate hygiene [14]. Implantoplasty consists of polishing the implant surface that has been uncovered by the loss of peri-implant bone height [15]. During IP there is a large release of metal particles into the peri-implant tissue as a result of a drilling process. After peri-implant surgery, often these metal particles cannot be completely removed from the bone and mucosal tissue around the implant. Several studies suggest that Ti particles released into the peri-implant tissue could trigger peri-implant bone loss [16,17,18]. Furthermore, Soto-Alvaredo et al. [19] found that Ti nanoparticles and ions exert a cytotoxic effect upon human enterocytes and murine osteoblasts. However, the impact of such metal debris released during IP on a living organism remains unclear.

Although IP has been used as a technique for mechanical decontamination of the dental implant with peri-implantitis, the physicochemical and mechanical properties, as well as the cytotoxicity of metal debris released during this procedure has not been addressed in detail. Thus, it is crucial to analyze these properties of the Ti6Al4V particles released during IP in order to understand the cellular and biological behavior of these particles [20]. Indeed, there are still important parameters to be determined, for example the specific surface area, nanoindentation or cytotoxic effects of such metal debris upon osteoblasts and fibroblasts.

The main objective of this in vitro study was to analyze the physicochemical properties of the metal debris released after IP of Ti alloy (Ti6Al4V) dental implants. In addition, as a secondary objective, the cytotoxicity of these metal particles was evaluated by assessing the cell viability of human fibroblasts and osteoblasts, as they are the main cells of the peri-implant tissues.

## 2. Materials and Methods

### 2.1. Sample Preparation and Collection

Implantoplasty of 60 Ti6Al4V dental implants (Avinent Implant System S.L., Santpedor, Spain) was carried out by the same investigator (J.T.-S.) following the drilling protocol described in the study published by Costa-Berenguer et al. [21]. Using a GENTLEsilence LUX 8000B turbine (KaVo Dental GmbH, Biberach an der Riß, Germany) under constant irrigation, the surface was sequentially modified with a fine-grained tungsten carbide bur (reference H379.314. 014 KOMET; GmbH & Co., KG, Lemgo, Germany), a coarse-grained silicon carbide polisher (order no. 9608.314.030 KOMET; GmbH & Co., KG, Lemgo, Germany) and a fine-grained silicon carbide polisher (order no. 9618.314.030 KOMET; GmbH & Co., KG, Lemgo, Germany). The samples were lyophilized to remove the water from the metal debris released during IP. Once 10 g of Ti6Al4V powder was obtained, the physicochemical characterization and cytotoxicity studies were carried out. Ti6Al4V studied was Extra Low Interstitial (ELI) being in atomic percentage: 5.9% Al, 3.9% V, Fe < 0.03%, C < 0.01%, O < 0.01%, Ti balance. The chemical analysis was realized by EDS microanalysis.

### 2.2. Specific Surface Area

The specific surface area analysis was carried out using ASAP 2020 equipment (Micromeritics, Norcross, GA, USA). Nitrogen was used as an adsorbate. The metal debris obtained was degassed at 100 °C under vacuum conditions (10 µm Hg). The specific surface area was analyzed by applying mathematical calculations described by the BET (Brunauer–Emmett–Teller) theory [22]. An analysis of the specific surface area was carried out in triplicate.

### 2.3. Granulometry

The particle size of the metal debris was measured using the Mastersizer 3000 (Malvern Panalytical, Malvern, UK). This unit uses the laser diffraction technique to measure particle size by measuring the intensity of scattered light as a laser beam passes through the sample of particles. This test was carried out in a wet medium, using ethanol as the liquid scattering medium, and with the necessary amount of metal debris to bring the scattering obscuration and the sample within the optimum range of 5% to 10%, which was finally adjusted to 7%. The particle size range that can be analyzed with this equipment is between 10 nm and 3.5 mm. Mechanical and ultrasonic agitation methods (2500 rpm and 50% sonication, respectively) were used to avoid possible agglomeration of the metal debris during the particle size test.

To perform the particle size test, the refractive index of the Ti to be analyzed must be specified. For this purpose, an X-ray diffraction test was carried out. The anatase phase value was chosen, with a refractive index of 2.51.

### 2.4. Scanning Electron Microscopy

The morphology of the obtained particles was evaluated by scanning the electron microscopy (SEM) using the Phenom XL Desktop SEM microscope (PhenomWorld, Eindhoven, The Netherland) with an accelerating voltage of 20 keV. The particles were placed on a conductive adhesive carbon tape in order to improve the images.

### 2.5. Ion Release

The release of metal ions from the sample into the medium was evaluated according to ISO 10993-12-2009 [23]. In accordance with the standard, a medium/powder ratio of 1 mL per 0.2 g of sample was used. A total of 3 samples (*n* = 3) were prepared for analysis (10 mL of medium and 2 g of metal debris per sample). The liquid medium used for ion release was Hank’s saline solution (Sigma-Aldrich, Co., Life Science, St. Louis, MO, USA), which, being certified and commercially available, ensures homogeneity. The liquid in contact with the metal debris was recovered and filtered through a filter of pore size 0.22 µm. For analysis, it was acidified with 2% nitric acid (HNO_3_ 69.99%, Suprapur, Merck, Darmstadt, Germany) to avoid precipitation of the metal ions prior to the measurement of their concentration by inductively coupled plasma emission mass spectrometry (ICP-MS).

The samples were extracted at 5 timepoints: 1, 3, 7, 14 days and 21 days, as in similar studies [24]. The samples were stored at 37 °C in an incubator oven and were shaken at 250 rpm with an inclination angle of 30° to avoid settling of the metal debris during the assay and to ensure continuous exposure of all particles of the powder samples to the medium.

The samples were analyzed by ICP-MS (Perkin Elmer Elan 6000, Perkin Elmer Inc., Waltham, MA, USA). This technique enables a. quantitative multi-elemental analysis with an accuracy of 1 ppt for 90% of the elements of the periodic table.

### 2.6. X-ray Diffraction

X-ray diffraction analysis was carried out with a diffractometer (XRD; D8 Advance; Bruker, Karlsruhe, Germany) equipped with a germanium crystal and a Cu K_α_ anode. The system was operated at 40 kV and 40 mA.

A total of 1 cm^3^ of metal debris was used for the analysis. Data were obtained between 10° and 60° with a step size of 0.02° and time of 1 s per step. Phase determination and identification was performed using DIFFRAC.EVA software (Bruker, Karlsruhe, Germany). X-ray diffraction allows for the determination of crystalline structures, as well as the phase of multiphase crystalline materials.

### 2.7. Tensiometry

Due to the characteristics of the sample, the possibility of analyzing the contact angle by liquid sorption measurements using a Wilhelmy balance has been investigated. Under normal conditions, with samples presenting a comparable capillary force between samples (a certain homogeneity in the powdered material), this technique allows calculation of the contact angle using the Washburn method [25]. According to this method, liquid is extracted through capillary action. The mass increase in the tube, which is suspended from a force sensor, is determined with respect to time during the measurement. If the bulk metal debris is considered to constitute a capillary bundle, then the process can be described by the Washburn equation as follows:(1)$$\frac{m^{2}}{t}=\frac{c\cdot \;\rho^{2}\cdot \sigma \cdot \cos\theta }{\eta }$$
*m* = mass; *t* = sorption time; *σ* = surface tension of the liquid; *c* = capillary constant of the powder; *ρ* = density of the liquid; *θ* = contact angle; *η* = viscosity of the liquid.

The constant c includes the number of microcapillaries and their mean radius; it also depends on the nature of the powder and that of the measuring tube.

Sorption measurements of Ti metal debris were performed with a Krüss Processor Tensiometer K100 (Krüss Scientific Instruments, Matthews, CA, USA) using a specific accessory for sorption tests. Milli-Q water was used as the wetting liquid.

### 2.8. Cytotoxicity Assay

The cytotoxicity of the sample was evaluated by indirect exposure determination according to ISO 10993-5 [26].

The cytotoxicity assays were performed in triplicate (*n* = 3), the samples studied being: test sample (Ti6Al4V metal debris), positive control (cells seeded directly onto the plate), and negative control: medium without cells.

The samples were handled aseptically throughout the assay. The cytotoxicity test consists of evaluating the percentage cell survival of a known cell line when exposed to the medium that has been in contact with a given material. In this case, an indirect contact cytotoxicity test was performed according to the guidelines specified in ISO 10993-5 “Biological evaluation of medical devices”, part 5 “In vitro cytotoxicity tests”. To quantify cytotoxicity, the cell survival rate, which indicates cytotoxicity if <70%, was calculated.

Since these metal particles are in contact with both bone and soft tissue, two human cell lines were used: SAOS-2 osteoblastic (ATCC^®^ HTB-85) and HFF-1 fibroblastic cells (ATCC^®^ SCRC-1041). The cells were stored with dimethyl sulfoxide as cryopreservative at −180 °C and were assayed bimonthly for the absence of mycoplasma.

Cells were cultured in a humidity-controlled incubator with 5% of CO_2_ supply. As recommended by the manufacturer, McCoy’s Medium (Thermo Fisher Scientific, Waltham, MA, USA) was used for SAOS-2 culture and Dulbecco’s Modified Eagle Medium (DMEM; Thermo Fisher Scientific, Waltham, MA, USA) supplemented with 10% Fetal Bovine Serum (FBS; Thermo Fisher Scientific, Waltham, MA, USA), 1% L-glutamine (Thermo Fisher Scientific, Waltham, MA, USA) and 1% penicillin/streptomycin (Thermo Fisher Scientific, Waltham, MA, USA) was used for HFF-1 culture. The medium was stored at 4 °C, and the supplements at −20 °C.

The extracts were assayed according to Section 8.2 of ISO 10993-5. To do so, the material was incubated in supplemented medium at a ratio of 1 mL per 0.2 g of sample, for 72 h at 37 °C. Cells were seeded at a density of 2–10^4^ cells/mL for 24 h before contact with the sample extracts.

Cells were incubated for 24 h with undiluted extract (the whole extract without dilution) and 1/2, 1/10, 1/100 and 1/1000 of diluted extract, using complete medium for dilutions. Cells were inspected for adhesion and morphology before and after contact with the extracts. Once the assay was completed, cells were lysed with Mammalian Protein Extraction Reagent (mPER), and cell viability was assessed as lactate dehydrogenase enzyme activity (LDH; Roche Applied Science, Penzberg, Germany). The viability was calculated according to the manufacturer’s recommendations, measuring absorbance at 492 nm.

### 2.9. Statistical Analysis

Specific surface area was carried out in triplicate. Data were recorded using a Microsoft Excel spreadsheet (Microsoft^®^, Redmond, Washington, DC, USA) and subsequently processed with the Stata 14 package (StataCorp^®^, College Station, TX, USA). Means and standard deviations were calculated, except for the granulometry test, where the mode and percentiles were used.

## 3. Results

### 3.1. Specific Surface Area

The samples of the metal particles showed specific surface area values ranging from 0.2 to 0.4 m^2^/g (Table 1). This range is explained by the dispersion of the sizes of the metal debris studied, as shown by the granulometry results.

### 3.2. Granulometry

As depicted in Figure 1, the particle distribution exhibited a Gaussian distribution, typical of friction and machining processes. The equivalent particle diameter was 159 µm (range 6–1850 µm) (Table 2).

### 3.3. Scanning Electron Microscopy

The morphology of the particles was irregular, exhibiting different particle sizes. Scanning electron microscopy and X-ray energy dispersive microanalysis evidenced no impurities within the Ti6Al4V metal debris, as well as the absence of any traces of abrasive material or iron coming from the wear of the burs. Although there might be small amounts of steel in the particles, they are undetectable with current means because of their scarcity. This apparent absence of contamination might be due to the relative hardness of martensitic stainless steel in comparison with titanium alloy, which keeps frictional damage to the steel and steel wear to a minimum.

The metal debris had a flattened geometry with a “flake” shape and clear signs of plastic deformation (Figure 2).

Finally, the micrographs exhibited slip bands, revealing the Widmanstatten plate microstructures of the Ti6Al4V alloy, indicating that the material absorbed a large deformation.

### 3.4. Ion Release

Figure 3 shows the release curves of titanium (Ti), aluminum (Al), and vanadium (V) ions into the liquid medium with Hank’s solution as a function of incubation time.

Comparatively, the release of Ti and Al ions was slower than that of V ions. Vanadium ions exhibited rapid release in the first 24 h, while between day 1 and day 7 the release rate slowed down, but then further increased between day 7 and day 21. The release of V ions did not stabilize with incubation time; in contrast, Al ions showed a fairly constant release rate between 0 days and 21 days. Finally, Ti ions showed a constant release between day 0 and day 7. Between day 7 and day 14 there was a slight increase in the release rate, and between day 14 and 21 it was seen to have stabilized.

Vanadium is an element that favors the beta phase and is the phase that corrodes most easily in the physiological environment. This is the reason for the high release of vanadium into the environment.

### 3.5. X-ray Diffraction

The diffraction spectrum of the analyzed metal debris is presented in Figure 4, plotting the number of beads (peak intensity) versus the diffraction angle (2θ).

The analysis of diffraction spectrum results using DIFFRAC.EVA software (Bruker, Karlsruhe, Germany) in conjunction with the international diffraction database “Joint Committee on Powder Diffraction Standards” revealed the presence of two phases: one corresponding to the Ti-Al alloy with reference (01-077-6855 Al3Ti17 Y0. 1) “Aluminum Titanium” and the other corresponding to beta phase of the Ti6Al4V alloy. This might reflect that the Ti6Al4V alloy studied presented two phases: the alpha phase with a majority in a percentage > 95% and the beta phase which was approximately 5%.

### 3.6. Tensiometry

There were major differences in the sorption profiles between samples (Figure 5). These differences were indicative of the heterogeneity in particle sizes, which can significantly alter the capillary constant of the powder between different samples. Therefore, the determination of the contact angle using this technique seems unreliable.

### 3.7. Citotoxicity

The results of the cytotoxicity assay expressed as percentage cell survival are illustrated in Figure 6. The cells were adherent to the substrate plate and presented the expected morphology, both before and after incubation with the extracts.

The sample under evaluation exhibited cytotoxic effects in both the osteoblastic and fibroblast cell assays in the whole extract (undiluted). In contrast, it showed no cytotoxic effects in the diluted extracts (1/2, 1/10, 1/100 and 1/1000). In these solutions, the results do not present statistical differences significance with *p* < 0.1. Osteoblasts and fibroblasts are the main cells in the peri-implant tissues. The loss of cytocompatibility is a bad indicator since it suggests that the main cells of the peri-implant environment may be compromised.

## 4. Discussion

The results of the physicochemical characterization indicate that the metal debris released by implantoplasty exhibits different sizes and shapes and releases a greater amount of V ions than Ti and Al ions. Furthermore, the cell viability assays showed this metal debris to produce a significant loss of cytocompatibility in fibroblastic and osteoblastic cells. However, as limitations, it must be pointed out that the analyses were performed on grade V dental implants, specifically on a Ti6Al4V alloy. Therefore, further studies are required to reproduce our methodology using pure Ti dental implants or other types of alloys such as Ti-Zr. On the other hand, although a previously described and published milling protocol was used in the present study [21], numerous milling protocols have been described in the literature for IP [27]. Therefore, not all possible waste products that are released when the technique is modified have been studied. Nevertheless, it should be noted that our results suggest that the release of waste products from the milling burs was negligible.

Most of the current dental implants have a surface treatment to increase their roughness, thus increasing bone-implant contact and accelerating the osseointegration process [28]. However, these surface modifications also facilitate bacterial adhesion and deposition, causing biological complications, and moreover make it difficult to decontaminate the implant surface when necessary. Indeed, the surface roughness of implants seems to have an impact upon the progression of peri-implantitis and the size of the initial peri-implant lesion [29]. Implantoplasty has been proposed in order to reduce this bacterial colonization, since it eliminates the threads (macrosurface) and reduces surface roughness [30].

Several mechanisms of release of Ti particles and ions have been described in dental implants: friction of the dental implant with the bone during placement, mechanical wear caused by surface decontamination, corrosion caused by chemical agents or substance released by bacteria or inflammatory cells [31,32,33]. Among the different decontamination methods, IP is the technique that generates the greatest amount of metal debris, much of which cannot be removed and remains within the soft tissues and bone after surgery [2]. The volume of metal particles released by implantoplasty depends on the amount of dental implant surface exposed to the oral environment, as well as the number of implants requiring IP [14,34,35]. A growing body of evidence suggests that Ti particles may influence bone remodeling and play an important role in the development of peri-implant marginal bone loss [16,31,32,36].

In the field of orthopedics and traumatology, total hip prostheses constitute a predictable rehabilitation treatment, with an 85% survival rate after a follow-up of 25 years [37]. Nevertheless, in the long term, the most common reason for failure is a periprosthetic osteolytic reaction caused by metal particles released during wear and produced by friction between the joint surfaces [38]. This condition is known as “metallosis” and can lead to aseptic loss of the prosthesis [39,40]. In this scenario, the tissue surrounding the prosthesis becomes filled with black particles resulting in large stains in the soft tissues, as well as in an increased presence of metal ions in the blood. Management implies the removal of the prosthesis, cleaning of the damaged area and placement—if possible—of a new prosthesis [38]. Although the exact immunopathological mechanisms underlying this phenomenon remain unclear, Obando-Pereda et al. [41] reported that macrophages in contact with detached Ti particles, secondary to joint prosthesis wear, expressed increased levels of proinflammatory cytokines (TNF-α, IL-1β, IL-6). These cytokines, as well as reactive oxygen species (ROS), lysosomal enzymes or activation of the complement cascade, play an important role in bone metabolism and in the long-term viability of joint replacements [37,40,42].

The specific surface area of the metal debris ranged from 0.2 to 0.4 m^2^/g. These ranges of values are normal due to the size dispersion observed in the SEM images. These metal particles had a plate-like shape and exhibited a Widmanstatten pattern, indicating that the material had absorbed a large deformation [5,43].

X-ray diffraction testing showed the metal debris to be composed of an alpha phase corresponding to Ti-Al and a beta phase. This beta phase affords good mechanical properties to Ti and is generated by the presence of V [44]. In our sample, the release of Ti and Al ions was low and stabilized over time. However, the release of V ions was greater and showed no signs of stabilization. This behavior of V has been observed in Orthodontics and Traumatology [2,3,39,45,46] and could be explained by its binding energy and solubility. The implications of this rapid release of V, without reaching saturation, remain unclear, since toxicity studies in the human body are not easy to carry out. Even so, the release of V ions is an undesired event, as it reflects degradation of the alloy. A recent systematic review reported that when Ti6Al4V is affected by corrosion or wear, the release of V and Al particles has a greater cytotoxic effect compared to commercially pure Ti [4]. However, the amounts of V ions in our sample were relatively small (parts per billion [ppb]).

In fact, Ti-6Al-7Nb alloy was introduced in orthopedics and traumatology because it was suggested that released V is toxic [3]. However, to date there is not enough clinical evidence to confirm that V is harmful; as a result, Ti6Al4V alloys continue to be used in both dentistry and traumatology. In this regard, it would be interesting to carry out a study of animal models to ascertain the targets of Ti, Al and V ions and also to study their effect with controlled doses.

The cell viability tests showed that in most dilutions the mean survival values were above 70%, which means that the metal debris was not cytotoxic. However, there was a significant loss of cytocompatibility in fibroblasts and osteoblasts in the non-diluted extract. Recently, Barrak et al. [20] carried out an in vitro experiment in which they demonstrated that the metabolic activity of human fibroblasts after exposure to metal debris from IP of Ti6Al4V implants decreased after day 10. This phenomenon did not occur when fibroblasts were exposed to Ti implants. Thus, although the evidence is scarce, clinicians should be aware of the material from which the implant is made before performing IP.

Finally, it is important to highlight that the immune system responds differently to different particle sizes of a given material [47]: large particles can be encapsulated by a fibrotic capsule to isolate them from the medium; medium-sized particles can be phagocytosed by macrophages and eliminated from the organism; small particles are not identified by the immune system [48]. Therefore, considering the size of the metal debris in our sample, all three responses can be expected.

## 5. Conclusions

Metal debris from implantoplasty presented an average equivalent diameter of 15 µm (range 6–1850 µm) and a specific surface area of 0.3 m^2^/g on average. The particles in the studied sample were of plate morphology and presented different sizes. Furthermore, the release of Ti and Al ions was low and became saturated after 21 days, while the V ion release was considerably higher and did not seem to saturate during the study period.

Finally, the metal debris generated by implantoplasty produced a significant loss of cytocompatibility in fibroblastic and osteoblastic cells in the undiluted extract, which suggests that these metallic debris might damage the main cells of the peri-implant tissues.

## Figures and Tables

**Figure 1 materials-14-06507-f001:**
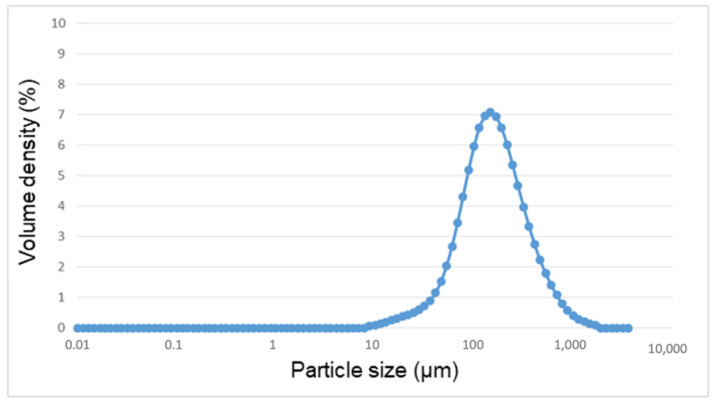
Distribution of particle size of the metal debris after implantoplasty.

**Figure 2 materials-14-06507-f002:**
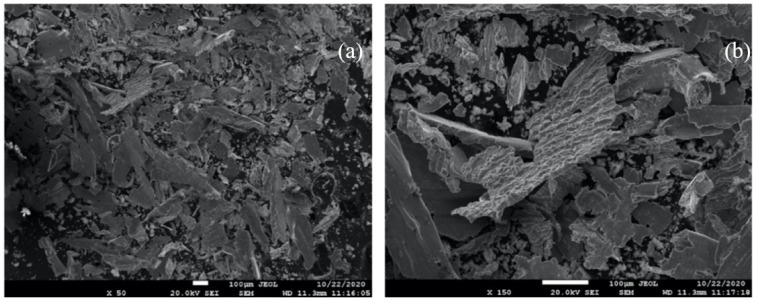
Scanning electron microscopy images under different magnifications; (**a**) ×50, (**b**) ×150.

**Figure 3 materials-14-06507-f003:**
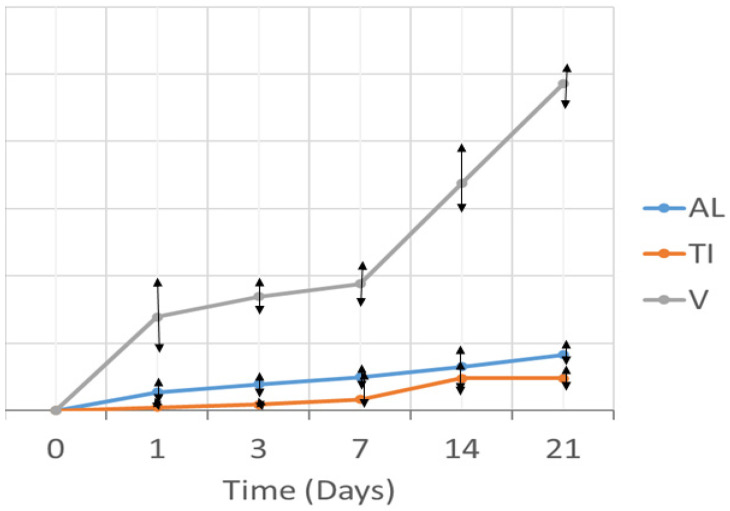
Titanium (Ti), Aluminum (Al), and Vanadium (V) ion release curve as a function of exposure time.

**Figure 4 materials-14-06507-f004:**
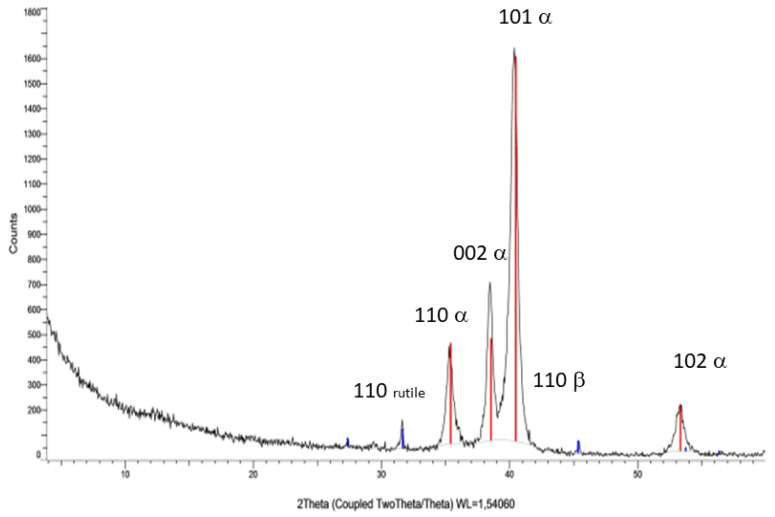
X-ray diffraction spectrum of the metal debris.

**Figure 5 materials-14-06507-f005:**
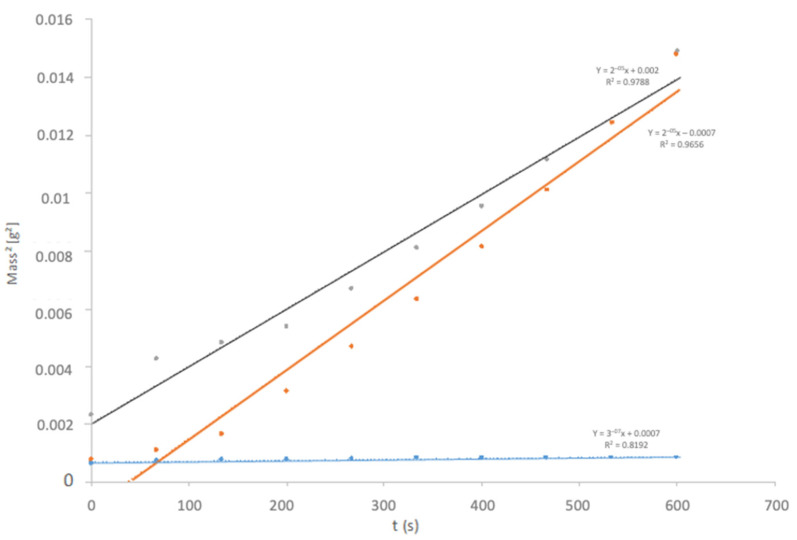
Representation of the sorption (mass increase as a function of time) of several samples of metal debris.

**Figure 6 materials-14-06507-f006:**
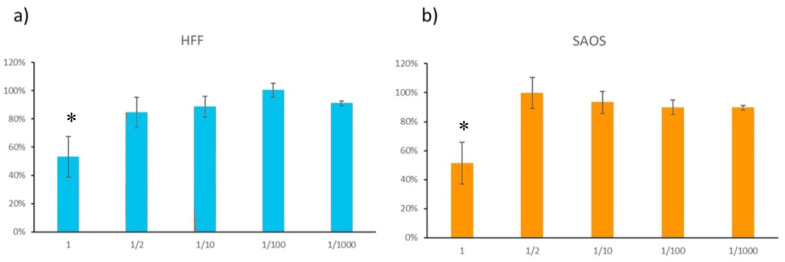
Cell viability in the cytotoxicity assay with HFF-1 (**a**) and SAOS-2 cells (**b**). * means that the sample had cytotoxic effect (loss of <70% of cell viability).

**Table 1 materials-14-06507-t001:** Results of the specific surface area of the sample of Ti6Al4V powder.

Assay	Specific Surface Area (m^2^/g) (SD)	C	Correlation Coeficient
1	0.3893 (0.0212)	23.21	0.9998
2	0.3550 (0.0316)	24.82	0.9997
3	0.2441 (0.0165)	22.07	0.9998

Abbreviations: C = BET constant, SD = standard deviation.

**Table 2 materials-14-06507-t002:** Granulometry results of the Ti6Al4V powder.

	Equivalent Diameter (µm)
Mode	152
10th percentile	61
50th percentile	159
90th percentile	433

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
