# Peer review of "Physicochemical and Biological Characterization of Ti6Al4V Particles Obtained by Implantoplasty: An In Vitro Study. Part I"

_materials, 2021, doi:10.3390/ma14216507_

Round 1

Reviewer 1 Report

- Add more current references (2020 – 2021).

-Improve the introduction with data about titanium alloys from the literature.

-Why you choose the Ti6Al4V alloy? Many studies show that long-term vanadium and aluminium can cause side effects.

-The authors do not make at all clear what the novel aspect of their work is, indeed their Introduction seems to indicate that everything they are looking at has already been done.

- Please discuss in Introduction about the applications for the Ti6Al4V alloy.

-Add a reference for ISO 10993-12-2009 used.

- Add a chemical composition of the alloy and explain how the EDS % composition values were obtained (spectra or area scan? number of spectra obtained to get the statistics?).

- Add on X-ray diffraction (Figure 4) the peaks (α or β peaks) and Miller indices.

- Generally the quality of the writing could be improved.

Author Response

Document attached

Author Response

document attached

Reviewer 3 Report

The article “Physicochemical and biological characterization of titanium alloy particles obtained by implantoplasty: An in vitro study. Part I” devoted to the problem of the metal debris released after implantoplasty of dental implants which can cause loss of peri-implant bone tissue. The theme of the article is very interesting and relevant. However, some comments should be providing.

  1. There is a repetition of the text “The release of vanadium ions … was greater than that of titanium and aluminum ions” in the Abstract.
  2. Particle morphology was not polyhedral since this assumes the presence of a three-dimensional polyhedra. In accordance with the micrographs shown in Figure 2, the particles had a plate-like shape.
  3. There is no mean deviation in the ion release graph (Fig. 3). It is possible that the concentrations of aluminum and titanium ions were approximately the same, since they varied within the experimental error.
  4. What is the reason for the different rate of release of vanadium ions during periods from 0 to 1, from 1 to 7 and from 7 to 21 days (Fig. 3)? How many samples were examined in parallel experiments? Maybe the rate of ions release will be approximately the same at all time periods, if the authors give the average deviation?
  5. It is extremely incorrect to assert about the presence of a phase "Yttrium Phosphide" in the Ti-6Al-4V alloy. Without a doubt, this is the beta phase of the Ti-6Al-4V alloy.
  6. If the data obtained by the method of Tensiometry (Fig. 5) are unreliable, what is the appropriateness to present them in the article?
  7. The Figure 6 does not show control as well as confidence level p ≤ x that was used for statistical data processing is not indicated.
  8. In the Discussion section, it is customary to discuss the results obtained in comparison with the literature data. In the presented work, in the discussion section, there are lengthy arguments that should be confirmed by specific results or not presented at all. For example, the authors describe the behavior of immune system when interacting with particles of different sizes. It seems that such research is needed in the work.

Author Response

document attached

Round 2

Reviewer 1 Report

English language and style are fine/minor spell check required

Author Response

Thank you for your remark. Authors have been reviewed the manuscript for English language spelling.

Reviewer 2 Report

The manuscript has been revised drastically.
But there is a question that needs to be added.

4. “The cell viability assays of osteoblasts and fibroblasts showed these particles produce a significant loss of cytocompatibility.” Is this good or bad? 

Osteoblasts and fibroblasts are the main cells in the peri-implant tissues. The loss of cytocompatibility is a bad new, since it suggests that the main cells of the peri-implant environment may be compromised.

It is recommended to add to the text.

Author Response

Thank you for your assistance. The abovementioned reference has been added in the abstract section and in discussion.